# The Non-Violent Liberation Theologies of Abraham Joshua Heschel and Mahatma Gandhi

Ephraim Meir [1,2]

1　Department of Jewish Philosophy, Faculty of Jewish Studies, Bar-Ilan University,
　　Ramat Gan 5290002, Israel; meir_ephraim@yahoo.com
2　Stellenbosch Institute for Advanced Study (STIAS), Wallenberg Research Centre at Stellenbosch University,
　　Stellenbosch 7600, South Africa

**Abstract:** This article explores how Gandhi and Heschel developed a liberation theology that was rooted in their religious praxis, which implied an active, non-violent struggle for the rights of the oppressed. A first section discusses what separates the two spiritual giants. A second section describes the affinities between them. The third, main section describes how they formulated a non-violent liberation theology that aims at the liberation of all.

**Keywords:** liberation theology; religions; suffering; tradition; Zionism; swaraj

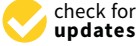



## 1. Introduction

Judaism and Hinduism have much in common. Studies on the relationship between both cultures have blossomed (Goodman 1994; Holdrege 1995; Goshen-Gottstein 2016; Theodor and Kornberg-Greenberg 2018; Brill 2019). Barbara Holdrege, a comparative historian of religion, has characterized the Hindu and Jewish communities as non-missionary "embodied communities," with modes of bodily practice such as purity codes, sexual disciplines, dietary laws, and scriptures (Veda and Torah) (Holdrege 2013). Judaism and Hinduism value people and land, language (Sanskrit and Hebrew), and rituals and laws (dharmic injunctions and halakha). They are both an orthopraxis, concerned with right actions and performances. Given these commonalities, it is not surprising that Gandhi's relational thinking and Jewish dialogical philosophies also display similarities.[1]

In previous publications, I dealt with the similarities between Gandhi (1869–1948) and Levinas (1906–1995) and pointed to Buber's nearness to Gandhi, despite his criticism of the Mahatma in the thirties of the preceding century (Meir 2021a, 2021b). Another Jewish dialogical thinker close to Gandhi is Abraham Joshua Heschel (1907–1972). In the present article, I demonstrate how Gandhi's active non-violent attitude in view of a change in society runs parallel with Heschel's social and political activities. More specifically, they were engaged in the non-violent liberation of people. In a way, Heschel's multiple social and political activities throughout the sixties until his death in 1972 continue Gandhi's liberating work for the Indian indentured workers in South Africa and later for the decolonization and spiritual independence (*swaraj*) in India. The former's endeavor to change the negative Christian attitude toward Jews and his—at that time unusual—engagement in favor of the black citizens in America are in line with Gandhi's pacifist worldview, aiming at the transformation of society. Just as Gandhi interpreted *swaraj* as implying a transformation of society and of the self, Heschel envisioned a model society in Israel, according to a prophetic blueprint. Gandhi and Heschel opposed war. I argue that both spiritual icons developed a liberation theology that was rooted in their unconventional religious worldview, which implied a non-violent struggle for the rights of oppressed and humiliated human beings. Their theologies had political relevance and led to social action.

Uma Majmudar has explored Gandhi's inner world and his search for Truth/God, which expressed itself in his active non-violence and service to others (Majmudar 2005).

She describes the evolution of Gandhi as a spiritual seeker and a man of vision and action. Heschel too combined religiosity and engagement in the world. Like Gandhi, he criticized a kind of religion that is severed from the world. Gandhi searched for the Truth in *samsara*, in the world. *Moksha* (liberation) was not outside, but inside, the world (Majmudar 2005, pp. 186–87). The two belonged to different worlds, but they underwent an amazing self-transformation, and their thoughts and acts had a global impact. They were profoundly religious men who were involved in a permanent struggle for equality and peace and who imagined a different society. Dr. Martin Luther King Jr. called Heschel a great prophet and was inspired by Gandhi. Reinhold Niebuhr, who taught at Union Theological Seminary in Manhattan, predicted that Heschel was "an authoritative voice not only in the Jewish community, but in the religious life of America" (New York Herald Tribune, 1 April 1951). For the pacifist clergyman John Haynes Holmes, Gandhi was "a great religious leader" and "the greatest man in the world" (Guha 2019, p. 184). Already in 1910 Pranjivan Mehta, before Rabindranath Tagore, called his friend Gandhi "Mahatma" "great soul" (Guha 2019, p. 164). The Trappist monk and later hermit Thomas Merton appreciated Gandhi and called him "one of the noblest men of our century" (Merton 1964, p. 32; 2007).[2]

Whereas Gandhi pled the cause of the Indians during his twenty-one-year stay in South Africa, Heschel supported equal opportunities for black people in the United States and defended the rights of three million Jews in the ex-Soviet Union (Kaplan 2007, pp. 214–34). They fasted and prayed as tactics of spiritual opposition (Kaplan 2007, pp. 305–13). Heschel opposed the American warfare in Vietnam and was the co-founder of the anti-war organization Clergy and Laymen Concerned about Vietnam. In a more radical way, Gandhi disapproved of war as such.

A related salient characteristic of Heschel's and Gandhi's worldview is that they radically put themselves in the service of others. Echoing the prophetic empathy with the poor and with victims of oppression, Gandhi sided with his discriminated and unfairly treated Indian brothers and sisters. Much as Gandhi and his Jewish friend Hermann Kallenbach (1871–1945), Heschel heeded a prophetic call and became actively involved in social and political actions. He inspired Martin Luther King's vision, embodied in his speech "I have a Dream." In his own famous lecture "Religion and Race," held in Chicago in 1963, he stated: "The exodus began, but is far from having been completed. In fact, it was easier for the children of Israel to cross the Red Sea than for a Negro to cross certain university campuses" (Heschel 1967, p. 85).[3] All had to be liberated. Racial prejudice was "*an eye disease, a cancer of the soul*" (Heschel 1967, p. 87). Marching at the side of King from Selma to Alabama, Heschel felt his legs were praying. In a letter to King, dated 29 March 1965, he qualified their famous walk a few days before (on 21 March) as "a day of sanctification."[4] Public humiliation was for him a form of oppression, worse than physical injury or economic privation (Heschel 1967, p. 88).[5] An infringement of human rights was a spiritual danger to the Kingdom of God. Heschel lived his prophetic faith in sympathy with the divine care especially for the under-privileged. Gandhi, in turn, wanted to restore the dignity of the Indians in defiance of British imperialism. Both men bring to mind the prophet Amos, who called for the liberation of all (Amos 9:7). Throughout history, the Bible was interpreted in view of the justification of racism. With the Bible in one's hand, one defended racial oppression. In order to counter such a scriptural-underpinned racism, Heschel and King referred to Amos in their struggle for equality of all (Johnson 2020).

Gandhi's permanent and fierce struggle for the equality of all parallels Heschel's opposition to racism as incompatible with religion: "Religion and race. How can the two be uttered together? To act in the spirit of religion is to unite what lies apart, to remember that humanity as a whole is God's beloved child. To act in the spirit of race is to sunder, to slash, to dismember the flesh of living humanity. Is this the way to honor a father: to torture his child? How can we hear the word 'race' and feel no self-reproach? [ . . . ] racism is worse than idolatry. *Racism is satanic*, unmitigated evil" (Heschel 1967, pp. 85–86).

With all their differences, Gandhi and Heschel imagined an alternative reality. Their religious thoughts inspired them to formulate a liberation theology, in which the unselfish

service to others was focal. Before expounding my main argument, I pay attention in a short section to what separates these exceptional human beings, who lived and worked in different situations. In a second move, I explore what unites them. In the last part, I explore how, in very different life settings, both spiritual leaders endeavored to formulate a kind of non-violent liberation theology that aims at the liberation of all.

## 2. Different Worlds

Gandhi's religiosity differed greatly from that of Heschel. Although Christian interpreters of Gandhi have superimposed their belief onto Gandhi,[6] he has to be understood firstly from his Hindu religiosity, which was already shaped in his childhood and youth (Majmudar 2005, pp. 42–44). He was raised in Vaishnavism, with its belief in Lord Vishnu and its avatars Rama and Krishna. He was influenced by the *Krishna-bhakti* that teaches that God is accessible to all and by the *Rama-bhakti*, which breached narrow caste, class, and gender divisions. In the latter tradition, the saint-poet Tulsidas occupied a central place. His Hindu epic *Tulasi-Ramayana* was read in Gandhi's household. Gandhi's mother adhered to the Pranami faith, in which the *Bhagavata Purana* of Vaishnavism and the *Qur'an* were closely linked. Vaishnavism and Islam peacefully coexisted. His mother also inserted Jain practices in her religious life, such as hard vows, palate control, and *ahimsa* (active non-violence), which was a means to the Truth. She practiced the Jain self-purification and self-perfection through *asteya* (non-stealing), *sunrita* (non-greed), *brahmacharya* (abstinence), and *aparigraha* (non-possession). She lived the Jain principle of *anekantvada* (many-sidedness of reality), which involved the validity-claim of all judgments. Gandhi's parents practiced the Vaishnava, Pranami, and Jain traditions (Majmudar 2005, p. 62).

Gandhi was a *bhakta*, who had God permanently in mind. Majmudar rightly remarks: "Although Gandhi believed in *Advaita Vedanta* (non-dualistic) philosophy of Hinduism, he was still a *bhakta* at heart—a man of prayer and inner contemplation, whose every breath, thought, and action was rooted in and dedicated to God" (Majmudar 2005, p. 186). By his experiments, he desired to realize the Truth/God. His optimism stemmed from his profound belief that God was present everywhere: all had the potential for goodness and forgiveness (Majmudar 2005, pp. 228–29). In the *Isha Upanishad* he found the kernel of his religiosity: all is filled by Isha (God) and belongs to God alone. One may enjoy what is given by God, but in detachment (Majmudar 2005, pp. 235–36).

Heschel belonged to an entirely different world. He was raised as a Hasid and he lived and transmitted a profound Hasidic tradition. Through his parents, Heschel was steeped in Hasidism. On his father's side, he descended from Dov Baer (the Maggid) of Mezeritch and Abraham Joshua Heschel of Apt; on his mother's side, he descended from Levi Isaac of Berdichev. The Ba'al Shem Tov { XE "Baal Shem Tov" } (Rabbi Israel ben Eliezer) and the Kotzker { XE "Kotzker" } rebbe (Menachem Mendel Morgensztern of Kotzk) lived in his soul: the Ba'al Shem Tov represented love and compassion, but the force that called out to change the world was the prophetic voice of the Kotzker, with whom Heschel profoundly identified. Heschel's heart was with the Ba'al Shem Tov and his mind with the Kotzker (Heschel 1996b, p. xiv). While the former brought God near to the human beings, the latter challenged the view of the Ba'al Shem Tov. The Kotzker did not rest upon a glorious past but searched uncompromisingly and restlessly for the truth. He protested against mediocrity. Edward Kaplan, Heschel's biographer, { XE "Kaplan" } succinctly depicts Heschel's personality by writing how the two divergent teachers shaped Heschel's spiritual life: the optimistic Ba'al Shem Tov and the abrasive, judgmental rebbe of Kotzk { XE "Kotzk" }. Heschel was alert to the pervasiveness of evil and self-deception and awed by God's concern for humankind (Kaplan 2007, p. xi). The Ba'al Shem Tov and the Kotzker rebbe { XE "Kotzker rebbe" } were present in his personality as two figures, who generated a creative tension in his soul.

As a result of their different religious background, Heschel and Gandhi were almost opposites. Gandhi believed that human nature is intrinsically good. Brahman was present in all and had to be uncovered in reality through active non-violence. Heschel also saw

God mainly in the poor and oppressed, whose battles he fought. Yet, he sided with the Kotzker rebbe, who emphasized humans' problematic nature.

Dissimilar to Heschel, who emphasized the differences between religions, especially Judaism and Christianity, Gandhi was less interested in differences. The religion of non-violence was the only true religion, the ocean to which all religions flow. Human beings are not really separate. On the individual as well as on the collective level, separation and differences are an illusion (*Maya*). Gandhi largely overlooked differences.

At first sight, the worlds of Gandhi and Heschel greatly differ.[7]

Gandhi and Heschel travelled through completely different worlds: Heschel from Poland to Germany, to England, and to the USA and Gandhi from India to London and South Africa. Heschel went his own way by deciding to study in Vilna and Berlin and by adding social activities to his professorship. Gandhi stood alone against his caste elders, who disapproved of his plan to study in London, and returned as a barrister. Although they ostracized him and his family, Gandhi was resolute in shaping his own identity and becoming a lawyer and a perfect English gentleman (Majmudar 2005, pp. 75–76, 81). He became a bargainer, making many compromises, in his attempt to realize God through his non-violent way of living.

Differences manifest themselves on several other levels. Gandhi and his *satyagrahis* cheerfully went to jail. Pledging vows, they gladly took upon themselves self-suffering and self-purification as a transforming power, aiming at convincing the opponent of his wrong way. Self-punishment for justice characterized Gandhi's *satyagrahis*, who presented their vulnerable bodies to the violent oppressor, showing his inequity and the violence of his laws. With his self-penance and even self-sacrifice until death, if necessary, Gandhi aimed at persuading the adversary and transforming him. Heschel did not share Gandhi's radical self-suffering and self-purification. He lacked the self-austerity that characterized Gandhi, who lived a simple, self-sufficient life in his ashrams. Although he was self-disciplined and with self-restraint, he did not adopt Gandhi's renunciation and austere way of life. He did not take upon himself suffering in order to melt the heart of the opponent. Gandhi loved to be arrested and felt privileged to be removed forcibly as a result of his civil disobedience. He pled for the severest punishment and praised God to be worthy of his prison experience (Sarid and Bartolf 1997, pp. 50–51, 60). His ascetic, anti-hedonist tendencies and self-punishment, including even fasting to death, were a means of self-purification. Whereas an ascetic Gandhi took upon himself voluntary suffering and other austerities, Heschel used different strategies in his fight against violence and injustice.

Gandhi interpreted the Gita allegorically: it was a battle between positive and negative elements in the human being. He also read the Hebrew Bible allegorically. He followed the Christian traditional way of reading Paul, who wrote that the letter kills, whereas the spirit frees (2 Cor. 3:6). Heschel too wanted a spiritual reading of the Hebrew Bible, but unlike the traditional Christian reading of Paul, he remained with the letter that contained the spirit.[8]

Gandhi loved the Sermon on the Mount, where Jesus says that one has to pray for the enemies and love the ones who hate you. Heschel did not go so far as to love those who hate you. However, both found in their foundational religious texts the motivation for their social and political struggle against humiliation and discrimination.

## 3. Affinities

Although Gandhi's and Heschel's worlds differ greatly, they also meet. As I will explain in detail in the next section, religion as implying action and empathy with suffering people was central in their worldviews. To be human implied to suffer with and for others.[9] Heschel wrote much about rituals and prayer (Heschel 1939, 1954). Yet, he emphasized that they had to lead to action. Religion was not a sanctimonious bubble above reality and far removed from everyday life. Like Heschel, Gandhi distanced himself from the humbug of his religion (CWMG 1999, 11: 64–65). Gandhi was not a temple-goer and did not build temples in his ashrams. Heschel did regularly attend synagogue services.[10] Yet, both knew that God

was not to be found in temples but rather in the face of the oppressed, whose suffering had to be alleviated. Religion had to be brought into contact with economic, social, and political life. It pervaded the entire existence and was relevant also and foremost for politics. One was in contact with God not in seclusion but by working in a concrete, messy world. Gandhi opposed the discrimination of the *Dalit*. In his view, untouchability was irreligion. He could not conceive of politics as divorced from religion (Harijan, 10 February 1940).

Gandhi and Heschel did not fear the authorities when they protested and breached existing norms. In Butlers' terminology: they imagined an alternative social society in which all lives are grievable and equal (Butler 2020). Judith Brown pointedly remarked that Gandhi "was profoundly God-centered and man-oriented" (Brown 1989, p. 90). God was to be found in the face of the downtrodden and outcast. All had "sparks of the divine"; all were rooted in God and therefore interconnected (Majmudar 2005, p. 139). Similar to Heschel, Gandhi served God by caring for the poor and the afflicted. He was ready to die for the sake of the untouchables. "I would far rather that Hinduism died than that untouchability lived" (CWMG 1999, 51: 62). He loved the poorest of the poor, the *Dalit*, calling them *Harijans* (Hari = God; jana = people). Gandhi and Heschel revitalized and regenerated their tradition. Just as Gandhi saw Brahman in all, Heschel perceived God in every human being.

### 3.1. The Insufficiency of Human Language

Gandhi was aware that human words are inadequate to express the Divine. In Heschel's depth-theology, the Ineffable cannot be reduced to human concepts. Scriptures are a mixture between the Divine and the human. Following the Jewish tradition, Heschel emphasizes that the divine name is ineffable. "Our creed is, like music, a translation of the unutterable into a form of expression. The original is known to God alone" (Heschel 1951a, p. 167). Our words are allusive and hinting (Heschel 1951a, p. 16). Religious language is not denotative and unequivocal but connotative and polyvalent: "[ . . . ] poetry is to religion what analysis is to science, and it is certainly no accident that the Bible was not written *more geometrico* but in the language of poetry" (Heschel 1951a, p. 37).

Gandhi and Heschel developed similar ideas on God and human concepts. In Gandhi's view, God appears in the form in which one worships, which implied that one must allow that He appears in other forms to others too (Chatterjee 1983, p. 23). The Truth or God was experienced in multiple ways; it was above words. The concept of God was not God. Similarly, Heschel refused to reduce God to a human concept of God. He wrote: "to equate religion and God is idolatry" (Heschel 1996a, p. 243). He could not stand the idea that one's God is not the God of others.

### 3.2. Beyond the Confessional

Heschel's thoughts on depth theology come close to Gandhi's thoughts on "religion underlying all religions" (Gandhi 2009, p. 41). Just as Heschel's depth theology united people, Gandhi's belief in the divine presence in everybody made it possible to look to what unites more than to what separates. The "religion underlying all religions" was the pursuit of Truth: it was present in all religions, transcended them, and allowed to see the equality of all. The religious truths were relative; they were different sides of the Truth (Majmudar 2005, pp. 106–7). For Gandhi, the Truth was God. It was the praxis and social action that made the manifestation of Brahman in everybody and everything visible. For Heschel, God's Name was at stake in the struggle for the equality of all.

Heschel and Gandhi did not accept Christian exclusivity. Yet, they had many Christian friends. The English priest Charles Freer Andrews was Gandhi's close friend, who appreciated, but also criticized, the Mahatma, as real friends do. Gandhi visited the Trappists at Mariann Hill, where he met monks who were vegetarians, living in silence and chastity, performing manual labor like carpentering, shoemaking, and printing. King and Merton were amongst Heschel's friends. In his endeavor to change the Christian attitude towards

Jews, Heschel was in contact with Cardinal Augustin Bea and even went to Rome to talk with the Pope for the sake of the Jewish people.

### 3.3. Prayer, Prophecy, and Activism

For Gandhi and Heschel, prayer and activism went hand in hand. Their religiosity was intimately linked to justice, compassion, and reconciliation. Heschel perceived prayer as turning oneself to God and making God immanent. In prayer, one becomes aware of the divine presence and realizes that one is the object of God's concern. To pray was "to take notice of the wonder, to regain a sense of the mystery that animates all beings, the Divine margin in all attainments" (Heschel 1954, p. 5). Before the divine face, the usual becomes unusual, and daily life becomes wondrous. In the "spiritual ecstasy" of prayer, one does not leave the world but sees it in a different light (Heschel 1954, p. 17). Self-consciousness is replaced by self-surrender, which is not a mystical negation of the ego but rather a state where God becomes the center and where one perceives the world in the mirror of the holy (Heschel 1954, p. 7). Heschel wanted to become a *shivitti*, a living reminder of having God's face permanently before oneself.[11]

In an interview with Carl Stern, conducted on 4 February 1973, Heschel stated that his work on *The Prophets*, published in 1962, prepared him for his social action. He became involved in the plight of the Vietnamese people, precisely because of his spiritual discipline, which made him sensitive for the suffering of others, dismissed by many American citizens. His religious inwardness expressed itself in reverence of the human being, who was a divine image: God was present in the human being. In Heschel's theology of pathos, the prophets were spiritual radicals, who identified with God's care for humankind and who became socially and politically involved. The human being was "a disclosure of the divine, and all men are one in God's care for man. Many things on earth are precious, some are holy, humanity is holy of holies" (Heschel 1991, pp. 7–8). Heschel criticized routine prayer without *kavvana* (intention). Prayer without an ethical life was a lie: "Prayer and prejudice cannot dwell in the same heart. Worship without compassion is worse than self-deception, it is an abomination" (Heschel 1967, p. 87). The spiritual discipline of prayer educated the human being to live a life in the face of God.

## 4. Non-Violent Liberation Theologies

The previous section dealt with the many parallels between Gandhi's and Heschel's thoughts and acts. The most striking affinity between them lies in the conception and realization of a non-violent liberation theology, in which God is present in all his creatures. They aimed at transforming human beings into humble servants for each other. To be sure, Gandhi was less systematic than Heschel in shaping his religious thoughts, but, like Heschel, he uttered them in view of the liberation of all. Their theologies contested economic and political inequalities. The main objective of their religious thought was the improvement of the situation of the oppressed, the poor, and the disenfranchised. They did not close themselves in small community life and did not content themselves with rituals. Their religiosity forbade the humiliation of others and implied the mending of the world.

Whereas liberation theologies in the seventies of the preceding century and also today are at times associated with violent struggles and armed rebellion, Gandhi's and Heschel's liberation theologies were explicitly non-violent. It is not to be excluded and even plausible that Heschel heard about Gandhi's *satyagraha* through his friends Martin Luther King and Thomas Merton, who admired Gandhi. In his prophetic religiosity, which asks for the mending of the world and for activity in the social, economic, and political arenas, Gandhi defended the defenseless and supported the cause of the oppressed and the poor. His active non-violence wanted to uncover Brahman in all that lives. Many times, the Bhagavad Gita affirms the identical existence of God in all beings. So, for instance, in chapter 13:18: "The Supreme God exists identically in all beings" or in chapter 18:61: "God abides in the heart of all beings." King, Heschel, and Gandhi loved to march for the cause

of the disadvantaged and in support of civil rights. Their marches were a religious act, a non-violent testimony to the presence of God in all human beings.

In the US, where racial segregation was common, Heschel stated: "From the point of view of religious philosophy it is our duty to have regard and compassion for every man regardless of his moral merit. God's covenant is with all men, and we must never be oblivious of *the equality of the divine dignity* of all men. The image of God is in the criminal as well as in the saint" (Heschel 1967, p. 95). For him, "[t]he symbol of God is man, every man" (Heschel 1967, p. 95). The human being was created in the image of God and in His likeness: "Man, every man, must be treated with the honour due to a likeness representing the King of kings" (Heschel 1967, p. 95).

In his Yiddish poems, Heschel identified with God's concern for suffering people. Parallel to Gandhi, he developed a theocentric view in which the human being was considered to be "something transcendent in disguise" (Heschel 1951a, p. 47). Whereas Gandhi uncovered Brahman in all human beings, even the evil ones, Heschel in his neo-Hasidic view looked for the divine "sparks" in the souls of all (Heschel 1996a, p. 250).

Gandhi lived his relationships with people in interconnectedness. Like Heschel, he was convinced that only deep interaction with others could lead to a better future. Brahman was in everyone; one had to make efforts in order to become conscious that creation was nothing less than Brahman's self-multiplication. The unity of mankind unveiled the oneness of God: "I believe in absolute oneness of God and therefore also of humanity. What though we have many bodies? We have but one soul. The rays of the sun are many through refraction. But they have the same source" (CWMG 1999, 25: 199).

### 4.1. Suffering the Sufferance of Others

Much like Gandhi, whose religiosity manifested itself in the understanding of the pains and suffering of others, Heschel suffered with the sufferers and shared the hope of the dispossessed and the oppressed (Kaplan and Dresner 1998, p. 79). In a parallel way, Gandhi experienced the suffering of (human and non-human) others as his own suffering. His favorite hymn "The true Vaishnava" begins with the words, "He is a real Vaihsnava, who feels the suffering of others as his own suffering" (Chatterjee 1983, p. 27). He saw his God Rama "face to face in the starving millions of India" (Chatterjee 1983, p. 17). Meeting poor peasants and living with untouchables, he felt face to face with God. A first untouchable family entered in the Sabarmati ashram, which was founded in 1915. Like Swami Vivekananda, Gandhi saw God in the faces of the poor (Chatterjee 1983, pp. 178, 256). Similarly, in his Yiddish poem *Ikh und Du*, Heschel recognized God and himself in the bodies of millions "as if under millions of masks my face would lie hidden" (Even-Chen and Meir 2012, pp. 16–17).

Gandhi's empathy for the poor and the maltreated runs parallel with the Jewish prophets, who—according to Heschel—identified with the divine pathos (Heschel 1962). Heschel and Gandhi were modern prophets, who cared for the humiliated and protested against white privilege and white supremacy. They felt the pain of others. Gandhi even wanted to be reborn as an untouchable, sharing their degradation. He belonged to the class of the traders. Yet, Brahmans, warriors, traders, workers, and untouchables were all equal in that all had to perform humble and polluting tasks. In Gandhi's non-dualist religiosity, God was present in all of them. Following John Ruskin and Leo Tolstoy, Gandhi thought there was no social hierarchy and no distinction in status in the *varnas* (classes) (Markovits 2000, pp. 182–88).

### 4.2. Guilt

Gandhi and Heschel emphasized the responsibility of each individual for all. They both wrote and talked about responsibility and about guilt as its counterpart. February 1938, Heschel lectured before a public of Quakers (Kaplan and Dresner 1998, pp. 259–62). He reminds their leaders that human beings are in the likeness to the Creator. He further quotes the Ba'al Shem Tov, who said: "If a person sees something evil, he should know

that it is shown to him so that he may realize his own guilt—repent for what he has seen."
All had to repent, victims and evildoers. God was in exile, imprisoned in temples. Only
by abandoning indifference could one bring an end to the divine exile: "Perhaps we are
all now going into exile. It is our fate to live in exile, but He has said to those who suffer:
'I am with them in their oppression.' The Jewish teachers tell us: Wherever Israel had to
go into exile, the Eternal went with them. The divine consequence of human fate is for us
a warning and a hope" (Kaplan and Dresner 1998, pp. 261–62). God wanted the human
being; He went into exile with his *Shekhinah* (the divine Inhabitation). He suffers with the
fate of the world, until all is united by human beings. In difficult times, Heschel said that
all have to repent.

In a parallel manner, Gandhi used to fast when something went wrong and when
people did not behave as non-violent *satyagrahis*. He felt himself the guilty party (CWMG
1999, 21: 462–64, 481). Gandhi blamed himself for what befell the Indians in colonial Africa
and India. Faced with evil, both Heschel and Gandhi strived for self-improvement. They
turned inward in self-examination. In their attempt to change evil, they looked for ways to
counter the tide.

### 4.3. Use of Religious Sources

In their liberation theology, Gandhi and Heschel developed a non-violent hermeneu-
tics of their foundational religious sources. Just as the Gita was for Gandhi the book
par excellence, the Hebrew Bible was for Heschel the most holy book. The Bible had a
message for the world: "It would be an achievement of the first magnitude to reconstruct
the peculiar nature of Biblical thinking and to spell out its divergence from all other types
of thinking. It would open new perspectives for the understanding of moral, social and
religious issues and enrich the whole of our thinking. Biblical thinking may have a part to
play in shaping our philosophical views about the world" (Heschel 1976, p. 23, n. 8).

The Bible as well as the Gita were used for violent purposes. Many times, one read
the Bible in function of white supremacy and the privileges of whiteness. In the United
States, one turned to the Bible in order to justify slavery (Johnson 2020, pp. 41–46). In South
Africa, the Boers saw themselves as elected. They read the Bible in a racist way. Gandhi
emphasized non-violence in biblical literature and interpreted the Gita as describing the
inner battles of the human being. His allegorical interpretation of the Gita runs parallel
with King's peaceful use of the Bible. In his previously mentioned speech, "I have a dream,"
King referred to the prophet Amos in affirming: " [ . . . ] we will not be satisfied until justice
rolls down like waters and righteousness like a mighty stream" (Amos 5:24) (Johnson 2020,
p. 49).[12] Heschel identified with King's peaceful struggle for an egalitarian society and
participated in mass protests of the black people in America. In a prophetical manner, he
became intensely involved in worldly affairs. Racism and religion excluded each other
(Johnson 2020, p. 48).

In their struggle for human rights, Gandhi and Heschel used religious texts that linked
religiosity and politics. They participated in public assemblies that contested the status
quo and promoted the equality of all. They desired to change politics and bring it into
contact with spiritual realities. In Gandhi's *Hind Swaraj* (Gandhi 2009), home-rule or *swaraj*
(swa = self; raj = rule), was not merely presented as political independence; it was an
elevated spiritual reality.

### 4.4. Religions in the Service of Humankind

In his lecture "No Religion Is an Island," Heschel stated that the diversity of religions
was "the will of God" (Heschel 1996a, p. 244). According to Harold Kasimow, Heschel
was "a Jewish interreligious artist" who does not enter easily in the Christian categories of
inclusivism or pluralism. He deems that Heschel cherished his Judaism and was convinced
of the truth of his religion, but he saw that there was more than one way of serving God,
although traditions were not "*equally* valid" (Kasimow 2009, pp. 199–200).

Heschel wanted religions to be involved in the world, but he did not develop an interreligious theology, in which interaction between religions leads to mutual learning and criticism and in which one does not leave a dialogue without being changed. Rather than formulating an interactive theology, he underscored the common task of all religious people, especially of people belonging to the Abrahamitic religions, and he described conditions for an interreligious dialogue (Heschel 1996a, pp. 239–40). He met with Cardinals Bea and Willebrands and with Pope Paul VI and was in dialogue with Thomas Merton and Reinhold Niebuhr. Through encounters and dialogue with religious others, he contributed to a more positive approach to Jews and Judaism.

Heschel and Gandhi did not develop a full-fledged interreligious theology. Yet, their openness to religious others and their interaction with them represent the basis for the construction of such a theology. Whereas Heschel focused on Jewish–Christian relations, Gandhi's interest was much broader. He read books on other religions, including Zoroastrianism and Islam. Prayers from different traditions were an integral part of the routine in his ashram. He valued the great variety of religions and focused upon the Hindu–Muslim relation, in view of the necessity of their cooperation in India. The spinning wheel that was so crucial in the *swadeshi* movement had to appear on the flag together with the green and red colors that represented Islam and Hinduism (Kapoor 2017, p. 142).[13]

Gandhi's and Heschel's religiosity was praxis-oriented. Gandhi conducted a lifelong interreligious dialogue and endeavored to move religious others to a non-violent way of life. There is an evolution in his thoughts on other religions (Meir 2021b, pp. 2–3). Gandhi himself was conscious of his own evolution and maintained that his later opinions were decisive. Although he accentuated the differences between religions less, he knew about "trans-difference," in which there is unity but also a multitude of particularities. He dealt with all kinds of diversity: children, women, languages, and religions. He was a real pluralist, although one may also find inclusivist standpoints when it comes, for instance, to Buddhism and atheism.[14]

Like Heschel, Gandhi was an engaged human being, who inserted religion conceived as non-violence into politics and social life. In this manner, he influenced many people, including Nelson Mandela and Martin Luther King and entire social movements such as engaged Buddhism (King 2009, pp. 2, 11). His Tolstoy Farm, created in 1910 and located near Johannesburg, was an experiment; it was "a center of spiritual purification and penance for the final campaign" (Gandhi 1968, p. 239). There were Hindus, Muslims, Parsis, and Christians (Gandhi 1968, p. 219). Although Christians and Muslims in the ashram were used to eating meat, it was decided to stop eating meat because of religious others. This, of course, solved problems between religions and promoted *satyagraha* as a non-violent religion. On the occasion of the decision of abstaining from meat, Gandhi noted: "[ . . . ] where love is, there is God also" (Gandhi 1968, p. 220). At school in the ashram, children of various faiths learned together. Muslims read the Qur'an, Parsis the Avesta, and one read, of course, Hindu texts (Gandhi 1968, pp. 224–25). The aim of the education, in which Gandhi was actively involved, was the cultivation of a spirit of friendship and service (Gandhi 1968, pp. 224–25). At prayer time, they sang *bhajans* (hymns), sometimes there were readings from Ramayana or a book on Islam (Gandhi 1968, p. 229). When Muslims fasted, there was only one meal for non-Muslims, at the evening before sunset, then for Muslims after sunset and in the early morning. Such a solidarity and sensitivity are characteristic for Gandhi's view on interreligious cohabitation. Like Gandhi, but less infrequently, Heschel prayed with religious others, for instance with King.

In his "Constructive programme" addressed to the members of the Indian National Congress in 1941, Gandhi reminds the Congressmen of the concrete steps of his political philosophy (Gandhi 2009, pp. 169–80). In view of the betterment of the life of all citizens, he emphasizes the need of a "communal unity" or "an unbreakable unity" (Gandhi 2009, p. 170). Every Congressman should "represent in his own person Hindu, Muslim, Christian, Zoroastrian, Jew, etc., shortly, every Hindu and non-Hindu" and "have the same regard for the other faiths as he has for his own . . . " (Gandhi 2009, p. 170).

In her doctoral thesis, Nicola Christine Jolly remarked that Gandhi's religiosity allowed him to be in touch with religious others, with whom he became friends. His friendship with them was relational, web-like, which supplements other ways of dialogue (such as the academic one or the official institutional one) (Jolly 2012, pp. 312–13). In his ashram, prayers were inter-religious, and religions were equal. With time, he adopted a positive attitude to inter-caste marriage and inter-dining (Jolly 2012, p. 108).

Jolly describes Gandhi as a pluralist, who maintained that no religion can have a full grasp of the multidimensional Truth, since all religions are mere human responses to God. For Gandhi, a moral atheist like Gora was truly "religious" (Jolly 2012, p. 43). Gandhi was influenced by atheists, who fought against injustice of the Hindu orthodoxy (Jolly 2012, p. 110). His standpoint towards atheists was inclusive, rather than pluralist, given the fact that in his adagio "Truth is God," the term God is still present.

Gandhi was a *sanatani* Hindu, who believed in the Vedas, the Upanishads, and the Gita. However, these scriptures had to stand the test of reason (Majmudar 2005, p. 190). As I noted, his was most of all a religion of non-violence, which he found in all religions. He therefore could not stand the humiliation and the violence committed towards the untouchables.

Gandhi wanted to understand Christians from a Christian viewpoint and Muslims from a Muslim viewpoint. Yet, he himself looked at the Jewish tradition through Christian lenses (Meir 2021b, pp. 5–10). In principle, he respected the writings of religious others and wanted to interpret the writings of a religious tradition in the way the best minds of that tradition understood their own tradition. Basically, he would not dare to criticize these writings as an outsider, since he himself had experienced how outsiders of his own tradition had attacked it out of ignorance: "I have nowhere said that I believe literally in every word of the Koran, or for the matter of that of any scripture in the world. But it is no business of mine to criticize the scriptures of other faiths or to point out their defects. It is and should be, however, my privilege to proclaim and practice the truths that there may be in them. I may not, therefore, criticize, or condemn things in the Koran or the life of the Prophet that I cannot understand. But I welcome every opportunity to express my admiration for such aspects of his life as I have been able to appreciate and understand. As for things that present difficulties, I am content to see them through the eyes of devout Mussalman friends, while I try to understand them with the help of the writings of eminent Muslim expounders of Islam. It is only through such a reverential approach to faiths other than mine that I can realize the principle of equality of all religions. But it is both my right and duty to point out the defects in Hinduism in order to purify it and to keep it pure. But when non-Hindu critics set about criticizing Hinduism and cataloguing its faults they only blazon their own ignorance of Hinduism and their incapacity to regard it from the Hindu viewpoint. It distorts their vision and vitiates their judgement. Thus my own experience of the non-Hindu critics of Hinduism brings home to me my limitations and teaches me to be wary of launching on a criticism of Islam or Christianity and their founders" (CWMG 1999, 64: 332).

*4.5. Between Tradition and Renovation*

In the formulation of his liberation theology, Heschel differed from Gandhi in that he was less of a reformer than Gandhi. He was a traditional Jew who went to the roots of his tradition, in which the identification with the discriminated was the alley to the Divine. As a Jew who knew about oppression, he developed a solidarity with the oppressed. He distanced himself from a pious Hasidism and found in the prophets a religiosity that was socially and politically relevant. The prophets were his heroes, because they sympathized with the divine concern for the poor and the suffering. God was a responsive God, a most moved mover, suffering with the sufferers. The prophets identified with His pathos. Heschel's and Gandhi's liberation theologies were the result of their religious engagement in the world. They refused to remain passive in front of political injustice, but they abhorred armed rebellion.

Gandhi lived his Hindu tradition and loved other religious traditions in as far as they were committed to an ethical life and to communication and dialogue. Religion with a capital letter was instrumental to create a "new we," in view of the good for all (Kalsky 2014a, 2014b). All religions contained care and compassion for the other. Gandhi deemed that belonging is not merely belonging to the own group, it was a worldwide belonging to all: a universal brother- and sister-hood. However, he opposed the abolishment of the caste system and the principle of hereditary occupation as the kernel of this system. The hereditary principle was an eternal principle and to change it was to create disorder. He reformed Hinduism, but only partially. His traditional standpoint raises the question of how this structural violence was compatible with his non-violence. He was severely criticized by the *Dalit* dr. Bhimrao Ramji Ambedkar (1891–1956), who opposed the caste system in his book "Annihilation of Caste." That system was hierarchical and discriminatory and had to be abolished. Drawing consequences, Ambedkar converted to Buddhism (Anand 2014). He wanted the *Dalits* to have the same status and opportunities as everybody else. He accused Gandhi of speaking in different voices: one in English in *Young India*—later *Harijan*—and another in his publications in Gujarati language in *Navajivan*—later *Harijan Bandhu*. It was not enough to allow *Dalits* the entry in a temple and give them the name *Harijans*, "people of God," as Gandhi did in 1933. More had to be done. Ambedkar could not see Gandhi as representing the *Dalits*. Against Gandhi who refused isolation and fragmentation, he wanted a separate electorate for the *Dalits* in the Congress.[15]

The human rights activist Arundhati Roy has also severely criticized Gandhi.[16] In her introduction, "The Doctor and the Saint," to the critical edition of Ambedkar's book, she mentions that Gandhi distinguished between the Indian business men, the "passenger Indians," and the indentured laborers, who mostly belonged to lower castes. She quotes Gandhi, who wrote on these laborers that they did not have any moral or religious instruction worthy of the name (CWMG 1999, 1: 200). The indentured workers were different from his own group. Roy notes that Gandhi did not want Indian prisoners in the same jails as the Kaffirs (CWMG 1999, 9: 256–57). Moreover, Gandhi and the "passenger Indians" were loyal to the British and served in the war on their side. They belonged to the "Imperial Brotherhood" (CWMG 1999, 2: 421). Yet, recently, Coovadia criticized Roy, who did not grasp Gandhi's radicalism, his personal independence, and his rapid change. Coovadia notes that the hope of "Imperial Brotherhood" was given up with Gandhi's translation of Tolstoy's *Letter to a Hindu* into Gujarati and with *Hind Swaraj* (Gandhi 2009) (Coovadia 2020, pp. 59–60). Gandhi had a complex approach to his tradition, which he kept and reformed. The problem of traditions lies in shaping the collective self on the negative background of others. In the worst-case scenario, the own value is affirmed, whereas others are devaluated. In this way, inequality and discrimination are born or kept alive; the humiliation of the ones constitutes the greatness of the others. Gandhi was aware that traditions are threatened by fear and segregation of the other.[17] He deemed that one may overcome these negative tendencies in tradition by realizing universal brother- and sister-hood. He affirmed the equal worth and dignity of every human being in an inclusive, non-discriminating community.

Gandhi departed from the traditional understanding of Hindu terms. For instance, whereas *ahimsa* was traditionally defined as non-injury and non-killing, Gandhi added to it compassion or love: "Where there is no compassion, there is no *ahimsa*" (CWMG 1999, 40: 192). *Ahimsa* meant active love. This positive interpretation and broadening of the traditional term *ahimsa* allowed Gandhi to be actively involved in the world. He referred to Tolstoy, who equated non-violence with active love and who would have understood non-violence better than anyone in India (CWMG 1999, 37: 262; Parekh 1999, pp. 112–16, 119–20).

One should not underestimate the radical novelty of Gandhi's independent thinking and of his doctrine (Parekh 1999, pp. 120, 123). Yet, he retained the four *varnas* with their hereditary occupations. These occupations were universal—imparting knowledge, defend-

ing the defenseless, doing agricultural work and commerce, and performing physical labor. They regulated social relations and conduct. With time, the harmful restrictions of inter-dining and intermarriage were added. Exploring Gandhi's alternative to or reformulation of the Hindu tradition, Dieter Conrad argues that, in a gradual process, Gandhi did not maintain the inferiority or superiority among the different occupations. Gandhi wanted to realize the truth of metaphysical human equality. However, this did not lead him to a condemnation of the caste system, in which one follows one's ancestors' occupation. He rather maintained the four divisions, which he viewed as functional: all had to serve and all had to hold on to their primary environment, avoiding in this way competition. In this manner, Conrad posits that Gandhi maintained and purified Hinduism from within (Conrad 1999).

Conrad's view on Gandhi and the Hindu tradition is not shared by others. Ambedkar already argued that Gandhi's approach to the Hindu tradition was highly problematic: untouchables are regarded as non-Hindus, since they are excluded from the *varna* system: oppression was affirmed in the name of religion. In our times, Anantanand Rambachan deems that Gandhi's ideal understanding of caste is not without controversy and that it is rejected by prominent *Dalit* leaders. He considers that it is a problem to affirm Hindu identity on the negative background of untouchables, who are dehumanized and humiliated. The challenge lies in the creation of a vision of the tradition that affirms the dignity of all. In the Advaita tradition, Rambachan finds that *Brahman* is present in all that is created. All, including the D*alits*, embody the infinite. Rambachan contends: "*Braham* includes everyone; caste excludes" (Rambachan 2015, p. 177). The Hindu perception of the divine equality is the basis for *ahimsa* as principle of non-injury or—positively—as the praxis of compassion and justice. Gandhi said: "No scripture which labels a human being as inferior or untouchable because of his or her birth can command our allegiance; it is a denial of God and Truth which is God" (Fischer 1962, p. 252). The four classes (*varnas*) were indeed complementary in Gandhi's ethical view. Rambachan, however, deems that there is no necessary correlation between birth and one's qualification for a particular kind of work. "Scripture," he concludes, "is not authoritative if it reveals anything that is contradicted by the evidence of other valid sources of knowledge" (Rambachan 2015, p. 184). A hierarchical social system is refuted empirically and cannot be justified by appeal to scripture. The goal of the *dharma* is the attentiveness to the good of all beings, not solely to specific groups (Rambachan 2015, pp. 185, 196).

In Rambachan's liberation theology, Gandhi did not go far enough. Hindus must recognize the inhumanity in the non-egalitarian, exploitative, and oppressive caste system. Concessions to the disadvantaged while maintaining a hierarchical social system is not enough. Advaita requires that one sees the suffering of the other as one's own (Rambachan 2015, p. 195). There is no separation between the self and the other: *atman*, the Spirit, unites all. The two Sanskrit words for compassion are *karuna* (used by Buddhists) and *daya*. This is parallel with the Hebrew word *rahamim*, mercy, a feeling in the belly. In the Jewish sources, God himself is called *ha-rahaman*, the Merciful. Additionally, in Islam, God is the Merciful, *al-rahim*; He is merciful and loves human beings (Qur'an 5:54). In Hindu thinking, God is in everyone, and interdependence implies that the suffering of others is one's own suffering. Rambachan values Gandhi's thoughts but also criticizes him.

Andrews, who had worked in the slums of London, equally opposed the caste system (Meir 2017, pp. 86–88). He lambasted the Christian missionary detractors of the Hindu tradition. Through Hinduism, he found God in everything. He influenced Gandhi, who in turn was attacked by the orthodox for making too many concessions and reforms, foremost in his attitude towards the *Dalit*.

The above criticism of Gandhi, however, does not diminish his merit of having developed a liberation theology for his own subdued people as well as for the British imperialists. With his liberating thoughts, he reimagined his own tradition radically. In Gandhi's and Heschel's perspective, tradition should never be a burden; it should remain open to the future. To value the wisdom of tradition does not contrast with its permanent

evolution. Gandhi was in *sampradāya*, the flow of tradition, which aimed to improve humanity. Heschel too deemed that tradition should remain an inspiration, as is eminently expressed in an aphorism attributed to the Austrian composer Gustav Mahler (1860–1911): "Tradition is the preservation of fire and not the adoration of ashes" ("Tradition ist Bewahrung des Feuers und nicht Anbetung der Asche"). Raised in the spirit of great Hasidic masters, he continued the tradition in a novel way. He rediscovered the social critic of the prophets.[18] Gandhi and Heschel developed a non-conventional approach to their religious tradition.[19] They criticized empty ritual and reformulated their tradition in view of mending the world. This approach allowed them to work out a liberation theology that protested against inequality and discrimination in a non-violent way.

### 4.6. Transformation through Non-Violent Protest

Gandhi and Heschel wanted to transform the human being. It was not enough to analyze: one had to act and to change reality. For Gandhi, this meant that one had to be the change one wants to see. His *satyagraha* was a love-force, based upon non-violence (*ahimsa*) as a process and upon a Truth (*satya*) that is to be realized. It aimed at changing people by not perpetuating the circle of violence. Active non-violence would bring forth a new world, in which state violence, oppression, humiliation, conflicts, and wars could be avoided or at least diminished. Instead of being motivated by greed, people could be metamorphosed into a universal brother-and sister-hood. Gandhi wanted to convince the British in South Africa that they do not have to envy the Indian traders through discriminating laws. He tried to arouse in them their humanity. Through *satyagraha* he wanted to melt the heart of opponents. He was convinced that, in the end, Truth would vanquish.

Gandhi was not an absolute pacifist. He recruited Indians for the English army in WW1. Heschel resembles Gandhi in that neither of them was a full-blown pacifist (Heschel 2020, p. 33). However, he adopted a non-violent approach in opposing the war in Vietnam and in siding with Martin Luther King's civil right movement.[20]

Gandhi actively protested against the colonial policy of the British vis-à-vis Hindus. In the same vein, Heschel protested against state violence in the case of the Vietnam war.[21] Both Heschel and Gandhi came up against racial prejudices, called by Gandhi the "deep disease of color prejudice" (Majmudar 2005, p. 98). They fought against racism and for human rights. Gandhi came up for the humiliated and oppressed Indians in South Africa. Heschel defended the rights of the Jews in Europe and in the Soviet Union and marched with King for the rights of the black people in America, where racism was prevalent. Gandhi's and Heschel's spirituality uttered itself in their concrete commitments. Yet, Gandhi himself was not entirely free from racial prejudice. He has even been accused of being a racist, of discriminating the African blacks. In his careful chronological account of Gandhi's writings, Nishikant Kolge answers the question whether Gandhi was a racist (Kolge 2016, pp. 88–93). Gandhi demanded separate lavatories and separate food for Indian prisoners. Kolge deems that, in the case of the lavatories, it was not a question of color but of hygiene and sanitation. In the case of food, the Indians had to get the food they wanted. Nevertheless, Kolge writes, Gandhi "held some strong opinions about the African blacks" (Kolge 2016, p. 93). However, if one situates his utterances in their context, his concerns "were guided by political consideration" in order to obtain rights for the Indians. Kolge further notes that, generally, Gandhi was supportive of the cause of African blacks. Gandhi, he concludes, "was neither a champion of the anti-racist movement which aimed at complete eradication of racial prejudices nor a fanatical racist who always showed disdain for the South African blacks" (Kolge 2016, p. 93).

### 4.7. Zionism and Swaraj: Beyond Mere Nationalism

Heschel and Gandhi significantly differed on the subject of Zionism. Heschel was profoundly shocked by the murder of six million Jews, amongst whom were many members of his family. After the Shoa, Israel was extremely important for him in order to save Jews. It was also and foremost an occasion to realize the prophetic visions (Heschel 1974). In

the thirties, Gandhi was asked to support the Jews in Germany and in Palestine. He did not favor the Jewish presence in Palestine that e belonged to the Arabs. In Germany and Palestine, the Jews had to adopt *satyagraha*. Gandhi did not favor legitimate, active self-defense (Meir 2021b, pp. 5–10). He did not conceive of Judaism as a peoplehood and did not favor Zionism. To be Jewish meant for him to belong to a religious denomination rather than to an ethnic group.[22] He understood the Jewish yearning to return to Palestine, but Zionism with its leaning upon the British was problematic. The real Jerusalem was the spiritual one (CWMG 1999, 48: 106–7). Gandhi did not take into account that many Jews identify as Jews because they belong to the Jewish people, notwithstanding a minimal or non-existing commitment to the Jewish religion.[23] To be a Zionist is one way of identifying as a Jew. Of course, Judaism is not merely a question of descent, of having a heritable foundation. It is rather performative, pertaining to communal belonging. A purely ethnic approach becomes quickly exclusive. Yet, Gandhi failed to see that Jewishness has also an ethnic dimension and that Zionism is an expression of Judaism. Zion was a safe haven for persecuted Jews in the thirties, when Gandhi was asked to raise his authoritative voice in favor of the Jews, which he failed to do.

In his article "The Jews," which appeared in November 1938, Gandhi did not support the Jewish cause, not in Germany and not in Palestine. Instead, he advised them to adopt *satyagraha*. He did not counsel them to defend themselves.[24] Martin Buber and J. L. Magnes wrote to Gandhi that they disagreed with him. In Buber's mind, the situation of the Indians in South Africa was quite different from that of the Jews in dictatorial Germany. The Indians were discriminated against, whereas the Jews were persecuted, tortured, and murdered. The Indians had India. The Jews had no homeland. To be a Jew did not mean solely to belong to the Jewish religion. Judaism was more than a religion: it was linked to a people, who returned to their land. Magnes similarly doubted that the proposed *satyagraha* would work in Nazi Germany. It was an unrealizable ideal. His own pacifism was in a crisis, and he wrote about a necessary war. Magnes's and Buber's letters were left unanswered. According to Buber, Gandhi only replied in a postcard that he regretted he did not have the time to write a reply (Crane 2007, p. 48, n. 10). It is a pity that we do not have Gandhi's explicit reaction to these two men, who admired him but who disagreed with him at this crucial moment.

One may only speculate on what Heschel would have written to the Mahatma at the time. He was thirty-one years old when Gandhi published his article "The Jews." He had succeeded Buber in the Center for Jewish Adult Education (*Mittelstelle für jüdische Erwachsenenbildung*) and the *Jüdische Lehrhaus* in Frankfurt. At the end of October 1938, he was arrested by the Gestapo and deported to Poland. Two years before, he had published his book *Die Prophetie*, a revision of his doctoral dissertation, in which he expressed his view on the prophets, who had an intercourse with God and who were empathic with the divine ecstasy and the divine active concern for humankind. Heschel fled from Europe and finally reached the United States, where he continued to be worried for the survival and equal rights of Jews. Zion was for him a refuge, but it contained also a promise.

Heschel saw Israel as a haven for persecuted Jews all over the world, but he looked also to Israel with prophetic eyes, full of expectation and hope. He lived the Bible, as Gandhi lived the Gita. The biblical history was for him alive, and Jews wrote the chapters of the Bible (Heschel 1974, p. 49). In his *Israel: An Echo of Eternity*, he looked upon Jerusalem through the eyes of the prophets. His was a prophetic vision. Jerusalem was a promise: the promise of peace and God's presence. God had a vision of man, whom He created according to His image, but the resemblance to God's image had faded rapidly: "God had a vision of restoring the image of man. So He created a city in heaven and called it Jerusalem, hoping and praying that Jerusalem on earth may resemble Jerusalem in heaven. Jerusalem is a recalling, an insisting and a waiting for the answer to God's hope" (Heschel 1974, p. 32). He addresses Jerusalem: "For centuries we would tear our garments whenever we came into sight of your [Jerusalem's] ruins. In 1945 our souls were ruins, and our garments were tatters. There was nothing to tear. In Auschwitz and Dachau, in Bergen-Belsen and

Treblinka, they prayed at the end of Atonement Day, 'Next year in Jerusalem.' The next day they were asphyxiated in gas chambers. Those of us who were not asphyxiated continued to cling to Thee. 'Though he slay me, yet I will trust in him' (Job 13:15). We come to you, Jerusalem, to build your ruins, to mend our souls and to seek comfort for God and men. We, a people of orphans, have entered the walls to greet the widow, Jerusalem, and the widow is a bride again. She has taken hold of us, and we find ourselves again at the feet of the prophets. We are the harp, and David is playing" (Heschel 1974, p. 17).

In his writings, Heschel famously stressed the importance of time as against place. Nevertheless, the land of Israel was important for the Jewish people. Did the land have holiness in itself? Did it have the special status it had for the medieval philosopher Jehuda Halevi { XE "Halevi" }? Heschel was certainly influenced by Halevi's writings. However, he recalled the ancient rabbis, who discerned three aspects of holiness: the holiness of the Divine Name, the holiness of the Sabbath, and the holiness of the people of Israel.[25] The ancient rabbis did not mention the holiness of the land *in se*. The holiness of the land was derived from the holiness of the people of Israel. Heschel quotes a variety of sources in order to prove this position.[26] The land, he concludes in contrast to Jehuda Halevi, was not holy during the early times of Terah and the Patriarchs; it only became "sanctified" by the people of Israel (Heschel 1951b, pp. 81–82). In the land, the people had to measure themselves according to prophetic standards and therefore the Bible had remained a moral challenge for them (Kaplan 2007, p. 337).[27] Heschel did not have a territorialist view of the land of Israel. He embraced a biblical and prophetic perspective: a person had to deserve to live in the land. For Gandhi too, *swaraj* implied personal improvement and cooperation of all with all.

Heschel pleaded for Jewish–Muslim cooperation in 1967. In a chapter entitled "Jews, Christians, Arabs," he devotes some thoughts to "Arabs and Israel" (Heschel 1974, pp. 173–89). As enthusiastic as he was with Israel of 1967, he did not forget the Arab neighbors. He wrote that in our world light and shadow are mingled: there is no wheat without chaff, no vineyard without weeds, no roses without thorns. There is joy over the rebirth of Israel but also pain over the suffering and bitterness in the Middle East (Heschel 1974, p. 173). Heschel cleaves to the original, prophetic dream: "Israel reborn is bound to be a blessing to the Arab world, to play a major role in their renaissance. The Arabs and the Israelis must be brought into mutual dependence by the supply of each other's wants. There is no other way of counteracting the antagonism" (Heschel 1974, pp. 182–83). He dreamed about communications running from Haifa to Beirut and Damascus in the North, to Amman in the East, and to Cairo in the South and about economic cooperation in agricultural and industrial development that could lead to supranational arrangements like the ones of the European Community. Young Israelis and Arabs could join in a mutual discourse of learning; excessive sums previously devoted to security could be diverted to development projects (Heschel 1974, pp. 184–85). Heschel asked from the nations in the Middle East to drop their antagonisms and antipathies, their hatred and fear, and demanded of them that they start to think in terms of one family. "The alternative to peace is disaster. The choice is to live together or to perish together" (Heschel 1974, p. 186). He ends with the words: "The Arabs and the Jews in addition to having a common background and history, early contacts and a prolonged and fertile symbiosis during the Middle Ages, have also another affinity in common: a heritage of suffering and humiliation". With the revival of Israel and the resurgence of the Arab nations, one faces many problems, according to Heschel, but cooperation was a vital necessity and a blessing for both (Heschel 1974, pp. 188–89).

With his theology that strived for the liberation of all, Heschel demanded that Jews and Arabs come to a covenant of brothers (*berit ahim*). He asked the perennial Jewish question: what is required from us now, what is the *Halakha* (the law)? The *Halakha* was to make peace with the Arabs, and, with this, he followed his prophetic vision of *shalom*.[28] Heschel told the story of Rabbi Ben Zion Uziel, who posited himself between two fighting camps and proposed to make peace: "Make your peace with us and we shall make peace with you and together we shall enjoy God's blessings on this holy land." He addressed

the Arabs as "our dear cousins": "Our common father, Abraham, the father of Isaac and of Ishmael, when he saw that his nephew Lot was causing him trouble [ . . . ] said to him: 'Let there be no quarrel between me and you, and between your shepherds and my shepherds, for we are people like brothers.' We also say to you, this land can sustain all of us and provide for us in plenty. Let us then, stop fighting each other, for we too, are people like brothers" (Heschel 1974, pp. 177–78).

In a mystical mood, { XE "Halevi" } { XE "Kook" } Heschel felt the hidden light in *Erets Yisrael* (the land of Israel). After the Shoah, in which there was no divine intervention, he again felt the wings of the *Shekhina* in *Erets Yisrael*. Yet, he also felt the pain of lack of fulfillment. He sensed that the return of the people was due to divine intervention. However, the people of Israel had to return to God in *qedusha* (holiness).

In Pinchas Peli's interview with { XE "Peli" } Heschel, conducted in Hebrew and broadcasted on Israeli television in 1971, Heschel spoke as a man with vision, who wanted an exemplary life in Israel, inspired by the prophets. He saw the continuation of the biblical narrative in the land of Israel. Concomitantly, he was concerned about the situation of the Arabs (Heschel 1974, pp. 25–26). Although he was convinced that Judaism was intimately connected to a concrete people with a concrete land, he was not a territorialist.

Gandhi's view on Zionism was far removed from Heschel's approach, but they both had a staunch belief in communication and dialogue. These two towering spiritual men were convinced that their respective traditions were an enormous contribution to the betterment of humankind. Their liberation theology criticized a mere nationalism without inner transformation of the human being. Gandhi broadened the traditional meaning of *swaraj*. For him, it was never merely political; it was first of all an ethical program. Similarly, in Heschel's view, the State of Israel was a means in order to realize a just society.

## 5. Conclusions

Heschel and Gandhi protested without fear against unjust and discriminating laws. Before the famous march from Selma to Alabama, closely followed by the FBI, Heschel read Ps. 27 in a chapel: "The Lord is my light and my salvation; whom shall I fear?" (Heschel 1998, p. 134). Away from the disastrous and all devouring fire of violence, Heschel and Gandhi marched for justice and for the liberation of all, for the Vietnamese as well as for the Americans, for the Hindus as well as for the British. Gandhi did so in *satyagraha* and Heschel in prophetic sympathy with the divine pathos. They did not react to violence with violence. Gandhi referred to *ahimsa* as the praxis of not causing harm to living beings, while Heschel contrasted the unbearable human violence with the prophetic perception of the silent sigh (Heschel 1962, p. 9). They opposed killings and white supremacy and envisioned a more equal society. Their remarkable religiosity expressed itself in non-cooperation and in the moral battle against human suffering and humiliation. Heschel never met Gandhi, but their thoughts and actions were correlative. If a meeting between these two giants of the spirit would have taken place, they would have had much to say to each other as two profoundly religious persons, who inserted a humanist religiosity in economic, social, and political life.

**Funding:** No external funding.

**Institutional Review Board Statement:** Not applicable.

**Informed Consent Statement:** Not applicable.

**Data Availability Statement:** Not applicable.

**Conflicts of Interest:** The author declares no conflict of interest.

## Notes

2    For a characterization of Merton's and Heschel's religiosity: (Magid 1998).

3   Twenty-three years earlier, the entrepreneur Prafulla Chandra Ray compared Gandhi's salt march in 1930 to the exodus of the Israelites under Moses (Guha 2019, p. 342).

4   For an account of the theological affinities between King and Heschel: (Heschel 1998).

5   "In the Hebrew language one word denotes both crimes. 'Bloodshed', in Hebrew, is the word that denotes both murder and humiliation. The law demands: one should rather be killed than commit murder. Piety demands: one should rather commit suicide than offend a person publicly. It is, better, the Talmud insists, to throw oneself alive into a burning furnace than to humiliate a human being publicly" (Heschel 1967, p. 88).

6   According to Markovits, Sir Richard Attenborough in his film of 1982 contributed to the iconic understanding of Gandhi. The cineaste makes a parallel between Jesus and Gandhi. He de-Hinduized Gandhi and simplified a more complex person. Romain Rolland saw Gandhi as a Francis of Assisi. In a sermon of 1921, Rev. Holmes regarded him as a new Jesus (Markovits 2000, pp. 31–37, 46–49). Markovits's merit is that he looks for Gandhi beyond the icon. He gives more weight to the South African period in order to understand Gandhi and depicts him in his everyday action, a humorous and daily economizing person and a "genius of agitprop" (Markovits 2000, p. 171).

7   In their private lives, the differences between them become even more pronounced. Heschel was a devoted father and spouse, for whom God dwelled in the harmony of husband and wife (Heschel 1976, p. 95). In line with mainstream Judaism, Heschel did not consider celibacy as an ideal in order to achieve detachment of the physical world (see Kornberg Greenberg 2018, pp. 214–15). Gandhi vowed to remain a celibate at the age of thirty-seven. His vow of *brahmacharya* (God/Truth-conduct) was taken after the Zulu Rebellion in 1906. Through his abstinence he wanted to control his sexual energy and sublimate it into spiritual power (Majmudar 2005, p. 69). He was an authoritative and jealous husband and marital battles were not infrequent. The biographer Rambachandra Guha notes that "for all his empathy and concern for those outside his family, Gandhi was curiously blind for the pain of his own sons" (Guha 2019, p. 64). He desired to shape his family in his image. Later in his marriage he was sorry that he caused disharmony and he came to appreciate his wife's non-violent resistance (Majmudar 2005, pp. 117–18). Majmudar pointedly remarks: "'The Father of the Nation' could not be a father to his own sons" (Majmudar 2005, p. 193). Gandhi was a reformer and not a stubborn traditionalist who was opposed to changes (Majmudar 2005, p. 207). Nevertheless, as I will explain further, he maintained the traditional four major occupations, without hierarchy between them. Majmudar defines him as a "critical traditionalist," as was Swami Vivekananda (Majmudar 2005, pp. 20–21, 112). At times, he attracted the wrath of some orthodox Hindus. Heschel cannot be called a reformer. He was a scion of a Hasidic dynasty, deeply steeped in the Jewish tradition as well as in a prophetic Judaism that identified with God's care for all. However, he left the pious Hasidic community in Warsaw, of which he was destined to be the Rebbe, in order to become active in the wider world.

8   Paul disagreed with Jewish missionaries in Corinth, who wanted the non-Jewish Christians to keep the Jewish laws. He argued that, since they were not Jews, they were not obliged to keep the Jewish Law. Therefore, the letter killed, whereas the spirit freed.

9   As E. Levinas formulates it beautifully in his Talmudic lecture "Damages Due to Fire": "To be human is to suffer for the other, and even within one's own suffering, to suffer for the suffering my suffering imposes upon the other" (Levinas 1990, p. 188); "L'humanité, c'est le fait de souffrir pour l'autre, et, jusque dans sa propre souffrance, souffrir de la souffrance que ma souffrance impose à l'autre" (Levinas 1977, p. 167).

10  Heschel did not share Buber's anti-institutional Judaism, but, like Buber and Gandhi, he wanted to renew the human being (Buber 1932).

11  The *shivitti* is a plaque that reminds of Ps. 16:8: "I have set the Lord always before me".

12  The verse is engraved at King's memorial in Atlanta.

13  The *swadeshi* independence movement boycotted foreign goods and promoted Indian products, in order to protect the poor against colonial exploitation.

14  In *Hind Swaraj*, p. 104 he still writes that "[r]ank atheism cannot flourish" in India. Later on, Gandhi approached the atheist Goparaju Ramachandra Rao (nicknamed Gora; 1902–1975) as an anonymous believer (in Karl Rahner's terminology) (Jolly 2012, p. 313).

15  The differences between Ambedkar and Gandhi do not prevent Debjani Ganguly to situate both personalities together on the stage of world history and to understand them in complementary terms as adopting both—in Bhabha's phrasing—a "vernacular cosmopolitanism". Both brought their small narrative in dialogue with world-enveloping ones. From a world historical perspective—so Ganguly—India is not a footnote of British history (Ganguly 2007).

16  In her article on Gandhi, parson Walter follows Roy in her harsh criticism of Gandhi (Walter 2016).

17  In his Hindu liberation theology, Anantanand Rambachan canvasses how the oppressive caste system and the oppressed *Dalit* community came into being. From the Rig Veda (around 1000 BCE), there was a split between those of noble descent (*arya*) and those who lacked this descent. Around 800 BCE, the *varnas* (castes) came into being: *Brahmanas*, priests, *kstriya*, soldiers, *vaisyas*, merchants and farmers, participated in the Veda ritual. The *sudras*, laborers, were impure and segregated. By the time of Manu (ca. 150 BCE), it was believed that to be born in a certain caste was the result of a good or bad karma. Other groups were outside the system. By the period between 400 BCE and 400 CE untouchables had to live apart, they were not allowed to eat with others and not intermarry. After contact with an untouchable, a bath of purification was required. Today—so Rambachan—15 percent

of the population of India is "untouchable" and the discriminating and marginalizing phenomenon persists. Given this context, conversion of untouchables to other religions becomes attractive (Rambachan 2015, pp. 169–70).

[18] Susannah Heschel notes that, whereas Gerhard von Rad placed the Hebrew prophets in the context of ancient Near East traditions, contesting their uniqueness as brilliant religious personalities close to God and presenting a counter-balance to the priestly religion of cult, Heschel rediscovered the prophetic social critique (Heschel 1998, p. 132).

[19] In his article on Heschel and Merton, Magid develops a similar argument: Heschel and Merton criticized modernity as well as tradition. Delving deeply into their spiritual sources, they went beyond institutionalization and convention. They criticized tradition and used it in order to criticize modernity, without abandoning the world (Magid 1998, pp.115–16, 121).

[20] He fought antisemitism and other forms of racism. For Susannah Heschel, the claim that Esau will forever hate Jacob should be rejected (Heschel 2020, p. 35). Instead of essentializing Christianity and stating that anti-Semitism was inherent in Christianity, one should give a chance to different, more peaceful relations. Susannah's father deemed that transformation was possible.

[21] Lately, the word "self-defense" has become more and more problematized (Butler 2020). State violence continued in the Trump era, with police killings of Afro-Americans, a high number of imprisoned black people and the public appearance of white supremacists. Cell-phone videos of witnesses changed the game in the case of police officer Derek Chauvin, who killed Georges Floyd. Movements as "black lives matter" and "Say Her Name" as well as slogans like "I can't breathe" have drawn the attention to white power and white privilege at the expense of colored people. Ethicists protest against negative stereotypes in black or brown bodies and develop a non-violent protest against discrimination. Gandhi and Heschel preceded these moral protests of today.

[22] Leora Batnitzky argued that Judaism came to be seen as a religion in the 18th century (Batnitzky 2011, pp. 13–14).

[23] Jewishness can be defined in manifold ways (Imhoff 2020).

[24] Gandhi also wrote: "[ . . . ] there are not wanting men who do believe that complete non-violence means complete cessation of all activity. Not such, however, is my doctrine of non-violence. My business is to refrain from doing any violence myself, and to induce by persuasion and service as many of God's creatures as I can to join me in the belief and practice. But I would be untrue to my faith if I refused to assist in a just cause any men or measures that did not entirely coincide with the principle of non-violence" (CWMG 1999, 20: 165). Notwithstanding this, Gandhi gave the Jews the Ahithophel counsel to behave as *satyagrahis* in the thirties.

[25] *Yalkut Shimoni*, parashat va-etchanan, dalet-he.

[26] Mekhilta parashat Bo, 12:1; Edyot 8, 6; *Mishneh Torah*, Terumot 1, 5; Tosafot Zebahim 62a.

[27] { XE "Kaplan" } Kaplan writes that Heschel preferred eternal values to ideology.

[28] Sic Ze'ev Harvey in his lecture on "Heschel on the Jews and the Arabs in the land of Israel" (Hebrew) at the Jerusalem Schechter Institute of Jewish Studies in 2003.

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
