# Peer review of "The Non-Violent Liberation Theologies of Abraham Joshua Heschel and Mahatma Gandhi"

_religions, doi:10.3390/rel12100855_

Round 1

Reviewer 1 Report

This is a fascinating study, showing very clearly the different positions as well as the commonalities of Heschel and Gandhi. And the perspective of a non violent liberation theology, derived from the overlapping points of these approaches, allows a vision for contemporary theology. I am very much impressed and think, the article has the potential for profound impulses of a theology for all humankind, without forgetting the different religious, world views and political preconditions and implications.

Author Response

Dear reviewer, thanks for reading and valuing my article. 

Reviewer 2 Report

89: "incompatible with racism" should read "...religion". 
119: Majumar -> Majmudar (in other lines as well) 
132: The Ba’al Shem Tov and the Kotzker should at least once be called by their names. 
212: Which -> whose? 
222: "sparks” supposedly refers to Heschel (!?) but not clear in this line. 
304: "frequently" -> it would be better to say: "at times"
620s: Only attributed to Mahler 
792: A Chapter title could be introduced, like "Conclusion". 

It is sometimes very confusing, how the authour jumps from Gandhi to Heschel and back again. Not every citation is easily attributed to one of them. The structure of the argument and of the article could be much clearer. 

But on the whole, it is a great and unique contribution

Author Response

Dear reviewer, thanks a lot for critically reading my article. I took into account your very useful remarks. Thanks again for the review.
